# Storyboarding HIV Infected Young People’s Adherence to Antiretroviral Therapy in Lower- to Upper Middle-Income Countries: A New-Materialist Qualitative Evidence Synthesis

**DOI:** 10.3390/ijerph191811317

**Published:** 2022-09-08

**Authors:** Lynn A. Hendricks, Taryn Young, Susanna S. Van Wyk, Catharina Matheï, Karin Hannes

**Affiliations:** 1Centre for Evidence Based Health Care, Division of Epidemiology and Biostatistics, Stellenbosch University, Cape Town 3000, South Africa; 2Research Group SoMeTHin’K (Social, Methodological and Theoretical Innovation/Kreative), Faculty of Social Sciences, Katholieke Universiteit, 3000 Leuven, Belgium; 3Department of Public Health and Primary Care, Katholieke Universiteit, 3000 Leuven, Belgium

**Keywords:** new materialism, assemblage, storyboarding, HIV, adherence, antiretroviral therapy, young people, perinatal infection, qualitative evidence synthesis, biopsychosocial

## Abstract

Young people living with perinatal infections of Human Immunodeficiency Virus (YLPHIV) face a chronic disease, with treatment including adherence to lifelong antiretroviral treatment (ART). The aim of this QES was to explore adherence to ART for YLPHIV as an assemblage within the framework of the biopsychosocial model with a new materialist perspective. We searched up to November 2021 and followed the ENTREQ and Cochrane guidelines for QES. All screening, data extraction, and critical appraisal were done in duplicate. We analysed and interpreted the findings innovatively by creating images of meaning, a storyboard, and storylines. We then reported the findings in a first-person narrative story. We included 47 studies and identified 9 storylines. We found that treatment adherence has less to do with humans’ preferences, motivations, needs, and dispositions and more to do with how bodies, viruses, things, ideas, institutions, environments, social processes, and social structures assemble. This QES highlights that adherence to ART for YLPHIV is a multisensorial experience in a multi-agentic world. Future research into rethinking the linear and casual inferences we are accustomed to in evidence-based health care is needed if we are to adopt multidisciplinary approaches to address pressing issues such as adherence to ART.

## 1. Introduction

Human immunodeficiency virus (HIV) represents a global public health challenge, with approximately 79.3 million people infected with the HIV virus since the start of the epidemic [1,2]. Although the burden of the HIV epidemic continues to vary considerably between countries and regions, HIV prevalence is highest in Africa [1], with East and Southern Africa being the most affected [3]. ‘Adolescents (10 to 19 years) and young people (10 to 24 years), especially young women and young key populations continue to be disproportionately affected by HIV’ [1,3]. A key population with limited representation in the literature is youth living with perinatally acquired HIV (YLPHIV) [4,5,6,7]. A perinatal infection can be described as vertical transmission from mother to child while pregnant, through childbirth, or through breastfeeding. Before the advent of antiretroviral treatment (ART), approximately 50% of children with perinatal HIV infections were expected to die before the age of 2 years. However, the survival of perinatally infected HIV-positive children into adolescence is on the increase globally [8,9]. YLPHIV now face a chronic rather than a progressively fatal disease, with treatment including adherence to life-long ART. Although ART is biomedically lifesaving, surviving into adulthood introduces new complex and challenging structural and contextual realities—most of which are unchartered in the literature. YLPHIV have been reported to experience higher levels of suicide ideation and mental health concerns and reported different clinical and psychological experiences of using and accessing treatment for HIV as compared to adults or young people who are infected through drug use and sex [8,10].

When theorising the phenomena of medication adherence, factors across the individual, social, and health systems as well as political and physical-environmental dimensions must be considered. Several theories across the disciplines of psychology and medicine have been used to understand and explain the barriers and facilitators of adherence to ART. The social cognitive theory, the health belief model, the theory of reasoned action/planned behaviour, and the protection motivation theory offer a more comprehensive approach to understanding health behaviours. The influence of external influences and previous experiences and their modifying effect on decision making is included in the self-regulatory theory. Further to this, the trans-theoretical model posits specific stages that people move through until behaviour occurs. However, these theories are limited by their exclusion of social pressures and interpersonal relationships, and it is challenging for existing theories to explain fully the health behaviours related to long-term HIV care. Munro and colleagues [11] evaluated health behaviour theories that have been applied to the understanding of responses to the challenges of long-term HIV care and highlighted how the biomedical perspective, behavioural learning theory, and communication perspective had very limited capacity to explain health behaviours in the larger complex context of HIV care [11,12].

So far, theoretical conceptualisations and logic models used to frame adherence research have mainly been inspired by the popularised biopsychosocial (BPS) model [13]. It is one of the most used theoretical frameworks, and it has been of considerable utility to those researching health and illness. The BPS model proposes three dimensions of health care, namely, biological, psychological, and social-environmental, and looks at the interaction between them. The philosophy of the BPS model posits a way of understanding illness on multiple levels, from the molecular to societal [12]. Traditionally, researchers tend to either privilege individual accounts over social structures, or the actors at work resulting in outcomes in intervention research, or the role of policy and guidelines in the practice and implementation. Recognising the limitations of these theories, others [11,14,15] have aimed to develop and evaluate frameworks that are more relevant to HIV care. Holtzman and colleagues [14] evaluated the use of Anderson’s behavioural model in mapping barriers and facilitators to ART adherence and retention. Katz et al. [15] aimed to describe the effects of internalised and enacted stigma on ART adherence. Munro and colleagues [11] described a model describing adherence behaviour related to TB treatment and determined that structural, social, personal, and health service factors may have a bidirectional relationship, and those responses to changes in one domain may be filtered through aspects of another domain. 

A recent comprehensive review [5] proposed a conceptual model of linkage, adherence to ART, and retention in HIV care. This review found that ‘multiple influences may act on an individual over time and result in changes in engagement and adherence behaviour’ in HIV-positive people, including several external influences (representing the effects of politics, society, and health systems, living in poverty, unpredictable life events), social influences (including stigma, social support, and relationships), and internal influences (such as motivations, desires, self-efficacy, and acceptance of the diagnosis, feelings of responsibility to care for family and others). A key feature of the model is the ‘tipping point’, demonstrating how people move in and out of adherence to ART and retention in care as life events occur and when factors intersect. However, the dualism of human and non-human continues to exist when developing theories and interventions for health care decision making. 

The analytical dimensions included in the BPS model, one of the most used analytical frameworks in health care, suggest that researchers have seldom paid attention to ‘environmental’ or ‘material’ factors in studying complexity, nor did they consider them as agents with the capacity to act and influence our behaviour. This line of thinking is strongly aligned with new-materialism, which has become an umbrella term used to represent a range of theoretical perspectives that share the return to a focus on matter [16,17] and what comes to matter for health.

In order to open pathways of understanding, not limited to dualisms, there has been a theoretical turn towards matter, and this has been represented by a range of theoretical perspectives [16,17], including New Materialism (NM). New materialist theorists see the world and its contents as unfixed, unstable, and relational [18,19,20]. Both the physical and social have material effects in an ‘ever-changing world’ [21], and the focus is on the materiality of social production rather than social construction [19]. New materialists believe that the capacity for ‘agency’—the actions that produce the social world—extends beyond human actors to the non-human and inanimate [22,23,24].

New materialism offers an opportunity to rethink the BPS model. New materialism has foundations in both anthropology and quantum physics and is emerging as a theory in the understanding of health [25]. Braidotti and De Landa [22], independently of one another in the late 1990s, introduced new materialism as a radical rethink of dualisms. Rather than presenting the relationship between humans and non-humans as hierarchical or categorical, they promote a line of argument that brings them together in an assemblage, hereby adopting a more lateral ontological perspective on relations between different species and materialities. Barad [18] posits new materialism as promoting the inseparability or the blurred boundaries between human and non-human entities or matter. New materialism ‘centres attention firmly on materiality; not only the materiality of flesh but also all the other physical and biological matter with which bodies come into contact, along with the sociocultural constructs and discourses that also affect bodies materially’ [19]. Matter is conceptualised as an independent agent with the potential to influence humans because matter is fully entangled with who we are and what we study. When coming together in the assemblage [23,26], in the complex configuration with no causal or linear logic, we propose that adherence to ART, as a phenomenon, comes into existence. The aim of this QES was to explore adherence to ART as an assemblage of different agents intra-acting with each other within the framework of the BPS model with a new materialist perspective.

### Why Is It Important to Do the Current Review?

Young people are especially vulnerable in South Africa due to reasons such as economic and educational inequality [3]—in the midst of the disproportionate health inequalities among Black South African young people due to the legacy of Apartheid. The institutional system of Apartheid divided people based on skin colour and other phenotypical features. This was done through the severe and brutal oppression and segregation of indigenous people in South Africa, stripping them of all forms of power. Many of these indigenous people are still trying to escape the economic and socially oppressive chains of Apartheid. Perinatally infected HIV-positive young people have been reported to experience higher levels of suicide ideation, mental health concerns, and reported different clinical and psychological experiences of using and accessing treatment for HIV to young people who are infected through drug use and consensual sex [27]. These findings may be due to young people having had an extended period of living with HIV, exposure to parents and/or family members who had HIV or may have died of HIV (loss of a parent), mode of disclosure of the HIV status, long term engagement with clinical services, drug resistance, and longer opportunities for issues related to stigma. The limited research available on perinatally infected young people prioritises this key population. So as to provide support and achieve the global Sustainable Development Goal (SDG) [28] of ending the AIDS epidemic and advocating for fair and equal treatment for all, more in-depth, contextually relevant research must be conducted to optimise adherence to ART, access to care, and retention in care for young people growing up with HIV. In 2021, an overview of qualitative evidence syntheses [7] (QES) concluded that there was limited review level evidence on the understanding of adherence to ART and retention in care for perinatally infected young people between the ages of 16–25 years. For the last ten years, there has been a continued call to prioritise research efforts on health-related behaviours that put young people at risk for ART non-adherence and how HIV-positive young people can best be supported to adhere to their ART treatment [4,29]. There has been a targeted call to focus resources and interventions on women and girls. In a global effort, the United Nations Transforming our world: the 2030 agenda for sustainable development [30] advocates for ending the AIDS epidemic, the fair and equal treatment for all, and for more in-depth, contextually relevant, transdisciplinary research to promote the optimisation of adherence to ART for YLPHIV. 

Adherence to treatment is a life-saving measure for people living with HIV and ensures their quality of life. Therefore, the study of non-adherence to treatment is of particular importance and interest. Implications of non-adherence are far-reaching and affect the person and their families, as well as causing long-term implications for health systems. In quantitative research using designs such as randomised control trials or case-control studies, missing treatment doses can compromise the data quality and analysis of research [31]. As this paper focuses on the assemblage as related to new materialism, in a qualitative explorative way, we move away from the concepts of numbers and statistics. In studying the phenomenon of adherence, with its barriers and facilitators, contextual dynamics, experiences, and perceptions, we do not attempt to make any causal hypothesis. In this qualitative evidence synthesis, we focus on non-adherence to treatment not as a measure of missed data but rather as the inclusion of the voices, perspectives, and experiences of missing groups or persons [32], specifically those of YLPHIV, from the literature and the narrative.

A QES of the experiences of HIV-positive people of ART adherence in sub-Saharan Africa [5] found that the interaction across dimensions of the individual, social relationships, communities, health care systems, and the political environment influences adherence to ART. Interactions between the environment and human health have long been of concern to medicine, and these interactions remain relevant to public health, epidemiology, environmental health, and health protection [24,33]. However, often unexplored in HIV research is the agentic relationship between non-human (things, materials), other-than-human (natural, environmental characteristics) and human agents, or intra-action, and the consequential impact on adherence to ART. 

## 2. Methods and Materials

This reporting followed the conventions of the Enhancing Transparency in Reporting the Synthesis of Qualitative Research (ENTREQ) guideline [34] (see Appendix A). The review was registered on PROSPERO (PROSPERO Reg Number: CRD42020147421) in 2019. 

### 2.1. Conceptual Framework

In this review, we extend the BPS theory with a new-materialist lens to explore our study phenomenon of adherence to ART. Our conceptualisation of adherence to ART for YLPHIV incorporates the ideals of complexity science, integrating the dimensions of biological factors (genetics, side effects, symptoms, biological response), psychological factors (experiences, perceptions, knowledge, attitudes, knowledge, mental health), social factors (social support, relationships, culture, traditions, community), political factors (health care system, policies, political climate), with the addition of material factors (differing objects) as an assemblage in constant flux (see Figure 1). The model visually demonstrates the intra-action of dimensions of treatment adherence, with matter in the middle of things as an atomic structure bounded by time and space. The dimensions are all in equilibrium, with no factor being weighted more than the other. The boundaries cross and are in constant motion. Circular in fashion, the factors are not static and vary along the continuum of factors. The arrows on either side of the atom represent the moving, in a bidirectional fashion, in and out, of time and context. 

### 2.2. Criteria for Considering Studies for This Review

#### 2.2.1. Types of Studies

Studies using a qualitative study design and qualitative methods for data collection and analysis were considered for this review. Eligible qualitative study designs included ethnographies, process evaluations, case studies, and mixed methods studies containing qualitative data. Qualitative data collection methods included observations, interviews, and focus groups. Qualitative data analysis methods, including thematic analyses, narrative analyses, grounded theory, content analysis, and descriptive presentations of findings, were included in this synthesis. All quantitative studies, such as cohort, survey, cross-sectional, randomised control trials, experimental and intervention studies, and other systematic reviews, were excluded, as were purely descriptive papers such as opinion pieces and editorials.

#### 2.2.2. Context

Studies conducted in low- to upper middle-income countries as defined by the World Bank [35] were eligible for this review.

#### 2.2.3. Types of Participants

We included papers that focused on participants who were infected with HIV perinatally or before the age of 10 years. The inclusion criteria for age in years was 10 years to 25 years old. Studies that offered perspectives of others on the experiences of perinatally infected HIV-positive youth were also considered for inclusion, including those from healthcare workers who provide services to HIV-infected people, those working within ART provision services, caregivers of HIV-infected children and young people, family members, and peers.

#### 2.2.4. Phenomenon of Interest

The phenomenon of interest in this review is adherence to ART or HAART. Although existing literature has varying definitions of adherence, this review will define adherence as taking medication or treatment as required. Nonadherence will be defined as ever missing or refusing a dose of ART.

#### 2.2.5. Outcomes

The anticipated outcome relevant to this review is an interpretation that captures the dynamic flow of complex intra-action between individual, social, health-system, political, and material dimensions that influences adherence to ART or HAART for YLPHIV.

### 2.3. Search Methods for Identification of Studies

The literature searches were conducted for the period 1 January 2013 up to November 2021 to identify the most recent literature. We searched the databases MEDLINE (PubMed), Embase (Ovid), CINAHL (EBSCOHost), Africa-Wide (EBSCOHost), ProQuest, Social Science Citation Index (Web of Science), and PsycInfo (EBSCOHost). We searched for MeSH terms and free text terms related to ‘adherence’, ‘young people’, ‘qualitative research’, and ‘LMICs’. The full MEDLINE search strategy is available in Appendix A. It was adapted for other databases, and no language limiters were included. We searched the reference lists of included studies. We searched conference abstracts for additional primary studies in the International AIDS Society Online Resource Library (http://library.iasociety.org/GlobalSearch.aspx) (accessed on 1 November 2021) and Dissertations and theses: ProQuest—dissertations and theses (http://www.proquest.com/products-services/pqdtglobal.html) (accessed on 1 November 2021).

### 2.4. Selection of Studies

After deduplication, two review authors (LH and SVW) independently and in duplicate screened abstract and titles, followed by full-text screening for inclusion. The software Covidence was used to conduct study selection and screening. Disagreements were resolved through discussion; when consensus could not be reached, a third author was included in the discussion. Study authors were contacted for additional information as required. Studies that did not meet the inclusion criteria were excluded with reasons. 

### 2.5. Methodological Quality Assessment

The quality of each study was assessed by two authors (LH and SVW), independently and in duplicate, using the Critical Appraisal Skills Programme (CASP) checklist for qualitative studies [36]. For studies that included both quantitative and qualitative methods and data, only the qualitative methods section was assessed with this tool and included in the analysis. The 10-item appraisal tool considers three broad domains, namely, ‘Are the results of the study valid?’, ‘What are the results?’, and ‘Will the results help locally?’. The domains were assessed as: yes, no, or can’t tell. Discrepancies were resolved through discussion with a third author (KH). No studies were excluded because of poor methodological quality.

### 2.6. Data Extraction

We developed and used an electronic data extraction tool, using Microsoft EXCEL, to assist with data collection. Data extraction of study setting, study objective, sample characteristics, HIV cascade description, the study design, the study data collection technique, and the method of analysis were conducted by two review authors (LH and SVW or KH), independently and in duplicate. Themes and material markers were extracted and visualised into images of meaning using pen and paper.

### 2.7. Data Synthesis by Storyboarding

To evaluate the complex interaction within the assemblage of the phenomenon of adherence, and specifically, to explore new materialism in relation to adherence to ART, we introduced a novel approach of *Synthesis by Storyboarding*, that enabled us to do justice to the very idea of creating an assemblage of intra-acting elements. The analytical technique of *storyboarding*, which consisted of 9 steps, helped us to make sense of the data collected from primary research papers. 

As visualised in Figure 2, we used the following steps in our synthesis: 

1. We used the BPS model as an initial compass to match article content with existing analytical concepts and extended it with a new materialism dimension. 

2. We familiarised ourselves with included studies through iterative and diffractive reading and discussing the papers with one another. The interpretations were generated through a diffractive approach of reading and analysing ‘stuff’. We read and reread insights and ideas from one primary study to another. A diffractive approach to analysis resists the neat categorisation [37] of findings we have grown accustomed to.

3. Using NVivo, two authors (LH and SVW or KH) independently extracted codes. A material marker refers to a textual unit that describes a tangible object, place, or space. To identify the material markers, we searched the quotes of participants, their descriptions, and interpretations of the authors of the included studies. We categorised tangible things, namely, things that can be observed through the senses, as material markers. We kept a log of the markers as we read the studies and then matched them with the themes under which we categorised the codes. 

4. Visual images, called *images of meaning*, representing the material markers that we derived from the data, were created by LH. 

5. We placed the images of meaning, with text from the papers, as well as summative text in the first-person voice, onto a comprehensive storyboard. 

6. We used the storyboard to identify the storylines of the papers. Storylines captured both the essence and the specific details related to the youngster’s experiences. Emerging storylines were discussed between authors of the review. Through discussion, we revised potential storylines evolving from the data. This was an iterative process that took multiple conversations. 

7. All images and the storyboard were created through rough drawings and symbols that later can be created digitally. The drawings had several iterations based on discussions among the authors.

8. Next, we provided a first-person narrative account of the storyboard from which we voice the ‘story’ of adherence to ART for YLPHIV from a visual display that combined narrative interpretations with drawings using a new materialist lens. The voice emerges from the complex relation among things, environments, affects, bodies, discourses, texts, and theory, in dynamically shifting arrangements and re-arrangements [38]. This voice serves as a vehicle to express the many utterances that become spoken from collective assemblages of enunciation. Its boundaries were determined by what we found in the included studies. The actual storyboard created is an artistically inspired form of evidence synthesis that integrates ideas and concepts from the primary studies with material markers identified. In line with the new materialist theory, the synthesis represents the materiality of meaning. The point of view from which the story is told is the young person perinatally infected with HIV. We opted for a first-person narrative (I/me) to situate the different parts of the storyboard. It emphasises the specific stance from which the action and events are told. The use of third-person accounts (her/him/them/it) is limited to those fragments where other agents, human or non-human, enter the first person’s thoughts or feelings. 

9. Finally, we provide a reflection of our role as interpreters of the narrative storyline. To position ourselves in this QES, we reflected on our expertise and our identities and how our positionality contributed to the construction of the research process and findings. As described at the end of the manuscript, all authors have experience in review methods and health. Presenting from the North and Global South, we shared experiences of health systems and theories that cut across place and disciplines. This is the first exploration of the theory of new materialism and its application to ART adherence. We started our journey with reading and building together through the protocol phase up to the writing phase. There were times that were defaulted to linear ways of thinking as expected in the sciences, but through discussions, reading, note taking, and recording our conversations, we were able to piece together the assemblage. Our aim is to provide a synthesis that is transparent. Therefore, through our Appendix A and presenting a step-by-step guide of our analysis process, we maintained the methodological strengths of this review.
Figure 2Synthesising multi-modalities using storyboarding.
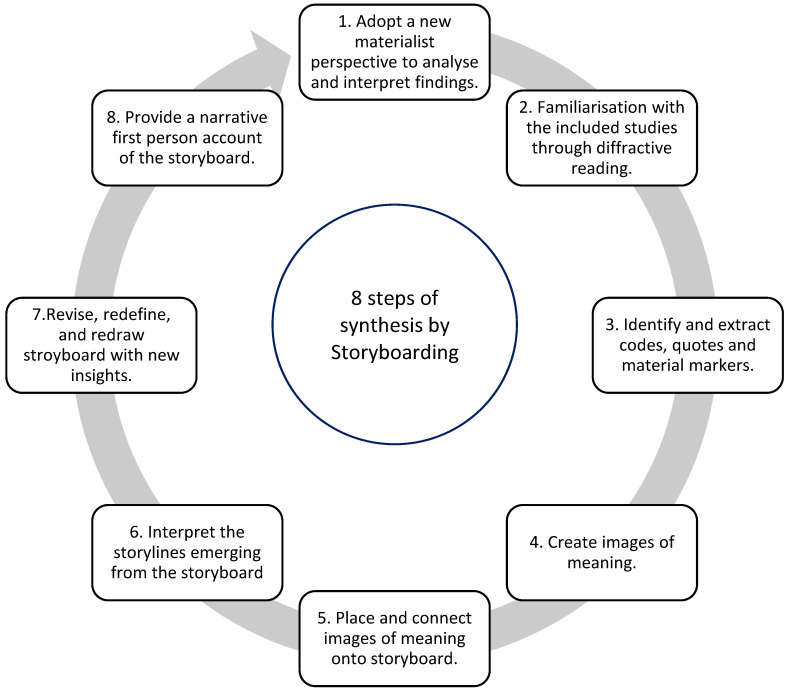


## 3. Results

### 3.1. Results of the Search

The database search retrieved 1030 records and is depicted in the PRISMA [39] flowchart below (Figure 3). Additionally, we searched the web and reference lists of included studies and identified 30 titles for screening. Therefore, our search yielded a total of 1060 records. We removed 63 duplicates in the database records we found and screened the titles and abstracts of 997 records. We excluded 883 records, and the full texts of 114 articles were retrieved. A total of 47 studies were included in the analysis of this review and 67 full-text articles were excluded with reasons.

### 3.2. Studies Included in This QES

Forty-seven studies met the inclusion criteria and were published between 2006 and 2021, with most included studies being published between 2015–2017 (See Figure 4). All studies included adolescents with perinatal infections of HIV. The age range for all YPLHIV was between 7–22 years (most with a mean age of 15 years). Some adolescents were orphaned, and others were double orphaned. Other participants included biological and non-biological caregivers, parents, siblings, aunts or uncles, teachers, grandmothers, and health care workers (doctors, nurses, counsellors, and pharmacists).

All studies explored adherence to ART or HAART for YPLHIV in low- to upper middle-income countries (see Figure 5). Nine studies were in Uganda [40,41,42,43,44,45,46,47,48], nine in South Africa [49,50,51,52,53,54,55,56,57], seven in Kenya [58,59,60,61,62,63,64], four in Botswana [4,65,66,67], three in Tanzania [68,69,70] and in Brazil [71,72,73], two studies in Zambia [74,75] and in Thailand [76,77], and one study each from Ethiopia [78], the Democratic Republic of Congo [79], the Dominican Republic [80], Ghana [10], Uganda-Harare-Zimbabwe [81], Rwanda [82], Uganda-Zimbabwe [83], and Zimbabwe [84]. All studies used qualitative research design, data collection, and data analysis methodologies. 

**Figure 4 ijerph-19-11317-f004:**
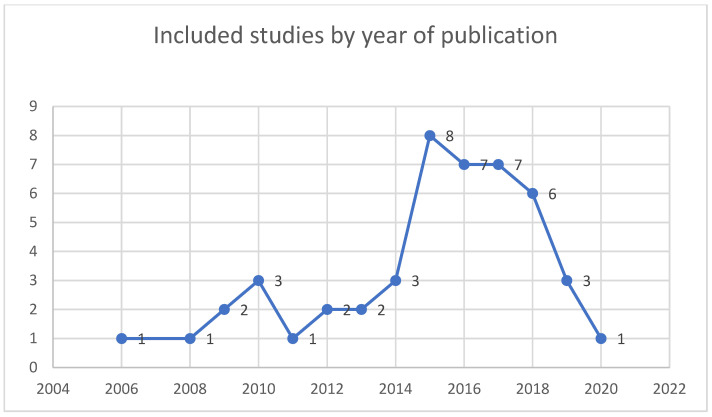
Included studies by year of publication.

**Figure 5 ijerph-19-11317-f005:**
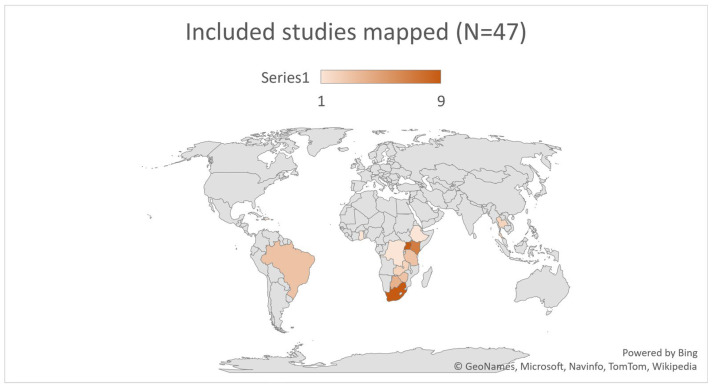
Included countries mapped.

A summary of the characteristics of studies included in the qualitative synthesis is presented in Appendix A.

### 3.3. Excluded Studies

One study could not be retrieved from the authors [85], and we excluded another 65 studies with reasons; 30 studies for wrong setting [86,87,88,89,90,91,92,93,94,95,96,97,98,99,100,101,102,103,104,105,106,107,108,109,110,111,112,113,114,115], 11 studies for the wrong phenomena of interest [29,48,55,116,117,118,119,120,121], 10 studies for the wrong patient population [117,122,123,124,125,126,127,128,129,130], 9 studies for wrong study designs [131,132,133,134,135,136,137,138,139], and 5 studies for the wrong outcomes [140,141,142,143,144]. 

### 3.4. Methodological Quality Assessment

Using the CASP checklist for qualitative research, we assessed most studies as having good methodological quality (see Appendix A). All studies clearly stated the aims of the research, and the research was considered valuable by our author team. Where there were not sufficient descriptions to make an assessment, we identified the item on the checklist as ‘can’t tell’. Although the qualitative methodology, designs, recruitment strategies, and data collection methods were appropriate and addressed the research issue, we found that most papers did not report or consider the relationship between the researcher and participants nor best practices regarding negotiating power dynamics within the research process, which may bias participants responses, thereby presenting uncertainty in the data. However, this may be a reporting issue and not necessarily an oversight by researchers to remain cognizant of their positional influence on the research process. All studies were approved by an ethics committee, and some had additional approvals from the health centre management and government gatekeepers. Studies differed in richness and sensitivity towards material and environmental dimensions.

### 3.5. Exploring Adherence with A New Materialist Lens—Synthesis by Storyboarding

#### 3.5.1. Theoretical Framework, Familiarisation, and Data Extraction

The themes identified from the primary studies were linked to various dimensions of the BPS model. For the biological dimension, these included changes in the body, poverty and food insecurity, biological sensations, and treatment failure. For the psychological dimension, this included motivation, hope, and resilience; internal stigma, depression, and suicidal ideation; fear of disclosure and secrets; and knowing and fearing loss. On the socio-cultural dimension, we agreed on romantic relationships and intimate partner violence, familial relationships, peer pressure and friendships, substance use and abuse, places in communities, and socio-economic context. We included a separate health system political dimension, a category with information on the location of the clinic, health care workers and consultations, the clinic, and the schools as themes (see Table 1). In line with our atomic model, material markers were positioned at the centre of things. 

#### 3.5.2. Creating the Images of Meaning, Interpreting Storylines, and the Storyboard

Reading the papers diffractively and drawing on visual images as means of data capturing onto a storyboard (Figure 6), we merged 9 storylines from the included papers into one holistic image (Table 1); these included: Navigating clinic visits, health care workers, and privacyDisclosure and psychological reactions to HIV diagnosisGrieving loss of caregivers and challenging familial relationshipsBodily changes, internal stigma, and suicidal ideationNegotiating power, sex, and risky behavioursFear of false judgement and stigma forces YLPHIV to keep their pills and status a secretWeighing up health against socio-economic constraints and educational prioritiesResilience, motivation, and future goals prompt adherence as habitual behaviourPlaces and spaces in the community that support and threaten adherence
Figure 6Storyboard of adherence to ART for YLPHIV.
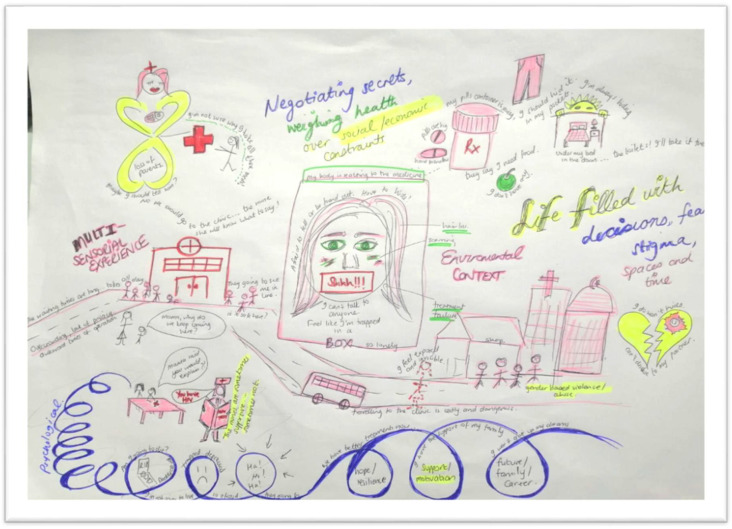


#### 3.5.3. The Story within the Assemblage of Adherence to ART

In what follows, we will voice the storyline from a first-person perspective. 

Storyline 1: Navigating Clinic Visits, Health Care Workers, and Privacy (Figure 7)

I went with my mom to the clinic today. I was planning on going to school, but I had to go with her instead. We left very early this morning. It was still dark. There was a long line of people waiting at the clinic before it opened. We have been coming to the clinic every few months and I see the doctor—I am not sure why. My mom gets nervous when people who we might know see us at the clinic. I know I’m supposed to not let them see me—I am not sure why. The clinic is so full. The smell of the clinic stays with me. When I take my tablets I think about my visit to the clinic, the nurses, and I see how much is left. When do I need to go there again?
Figure 7Navigating clinic visits, health care workers, and privacy.
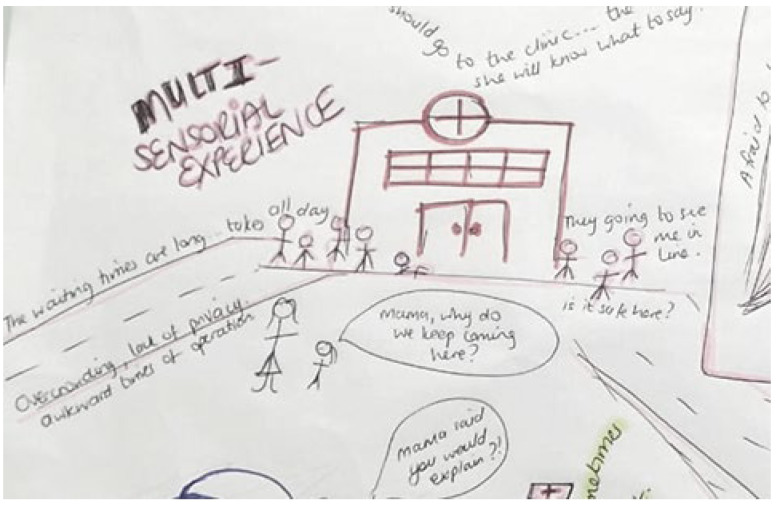


Storyline 2: Disclosure and Psychological Reactions to HIV Diagnosis (Figure 8)

Today was different. I had to take a test. The nurse pricked my finger and then put my blood onto a plastic board. I looked at my mom to explain but she looked at the nurse. The nurse had my file in her hand. I was wondering what was inside the file. She told me I have HIV. Am I going to die? My mom died. My friend...her dad died of AIDS. My mind is spinning, is this why I have been taking medicine? My mommy made sure that I took my medicine without anyone seeing. She knew they would tease me. I can’t tell my friends. Does this mean I cannot share food with them anymore? Or play with them? My carer does not let me keep my medicine in the fridge, because she says it will kill her children. I wish my parents were here to help me. I had thoughts of hanging myself...I was sad. My eyes were full of tears and then I felt encouraged by my doctor.
Figure 8Psychological reaction to positive HIV result.
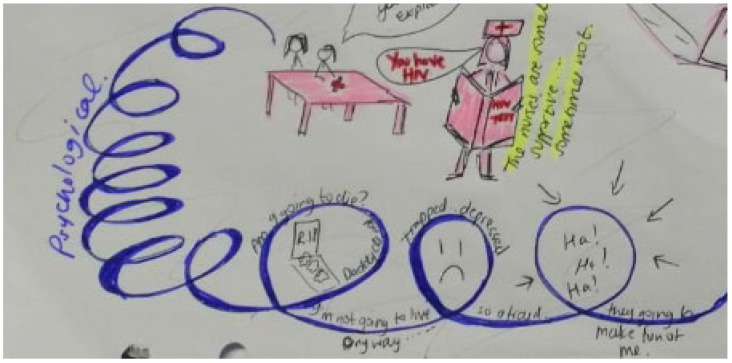


Storyline 3: Grieving Loss of Caregivers and Challenging Familial Relationships (Figure 9)

My mommy takes medicine all the time and sometimes she gets very sick. I wish I knew my dad, he passed away when I was very little. My friend lost both her parents, and she lives with her grandma. They keep giving me these big tablets to take, I do not know why I need to take them. Sometimes, they call it multivitamins and I must take it every day. I overheard my mom saying, ‘I do not know how to tell him, maybe I must take him to the clinic…the nurse will know how...’ What is she talking about? Since my dad died, we stay with my grandmother. She treats me differently to my siblings and cousins. I have more chores and she said that I do not need to go to school. I am in Grade 9. She says I should stay home and do chores because I am not like other children. What does she mean?
Figure 9Grieving loss of caregivers and challenging familial relationships.
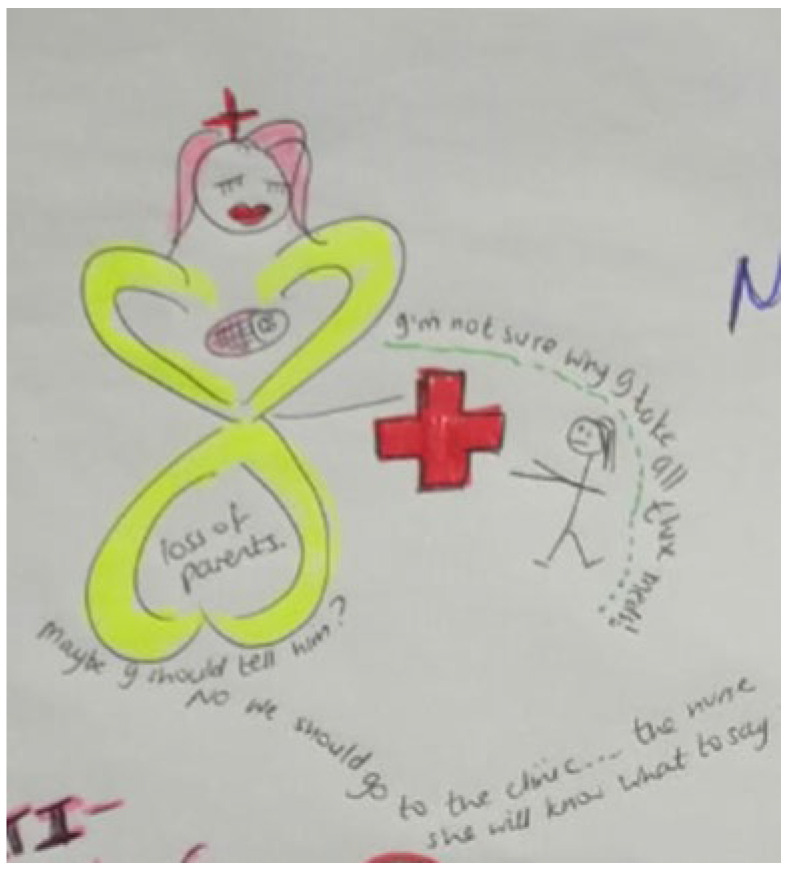


Storyline 4: Bodily Changes, Internal Stigma, and Suicidal Ideation (Figure 10)

The medicine has side effects, and it makes scars on my face. I am so lonely. I wish to take the knife on the counter or the rat poison and end things. It is not easy living here, there is still stigma, even I feel stigma inside, about myself. So many times, I had to change my medicine and get different types. Some of the pills made me tired and nauseas, others can make your hair fall out. I feel trapped in a box. I do not have anyone to talk to. I am afraid to tell anyone what I am going through. I must hide my medicine, when I go to the clinic, and I wear makeup to hide my scars. I have felt this way since I was young.
Figure 10Bodily changes, internal stigma, and suicidal ideation.
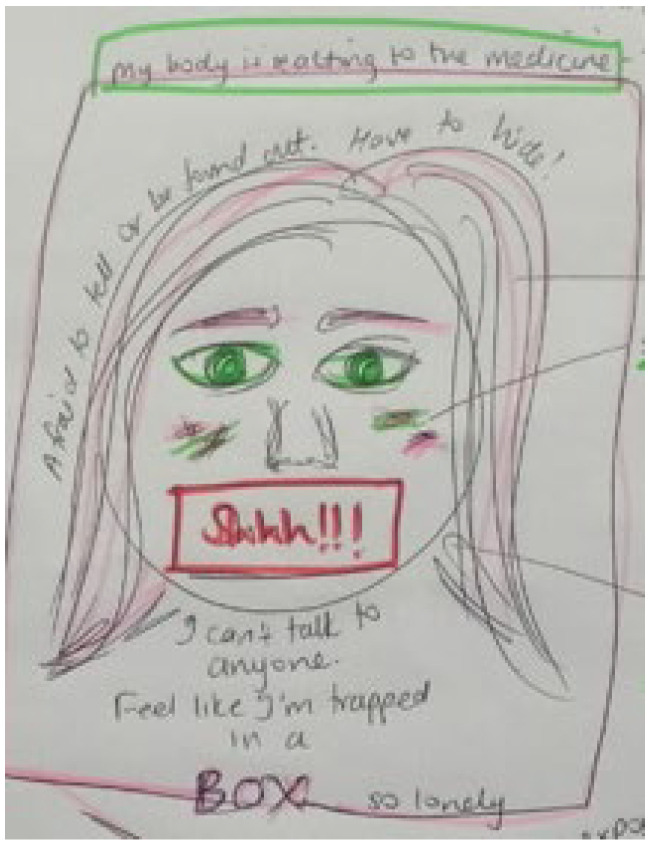


Storyline 5: Negotiating Power, Sex, and Risky Behaviours (Figure 11)

My partner does not want to use a condom but how do I tell them that I have HIV. It is not my fault I have this virus in my body. If I do not have unprotected sex, he will hit me with his hands. He will think I am using my body to cheat. My girlfriend always reminds me to take my medication. We have sex and she is not HIV positive. She looks after me when I am sick. If I drink alcohol, I will forget to wear a condom. My friends want me to go out with them and drink. I cannot tell them why I cannot drink. So, I drink, and tomorrow when I wake up in my bed, I will remember the things I was not supposed to do. Sometimes I do not want to remember that I have this virus. Other times I remember, and I know I must take my medication. I will tell myself I forgot to take my medication, but I just did not feel like it. I rather went out and socialised with my friends.
Figure 11Romantic partnerships.
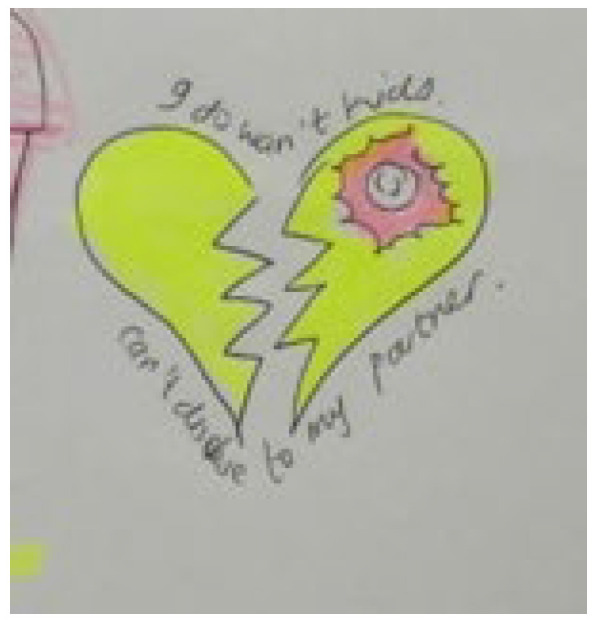


Storyline 6: Fear of False Judgement and Stigma Forces YLPHIV to Keep Their Pills and Status a Secret (Figure 12)

My pills are so big and hard to swallow. The container that they are in makes a lot of noise. It is hard to carry it around, so I wrap my pills in a tissue and put it in the pocket of my pants. That way no one will see me take it. I can also keep it in my handbag. At boarding school, I hide it in the back of my side table drawer or under my bed. I go to the bathroom when I need to take my medication. Sometimes I miss my clinic appointment because I do not want my friends to notice that I am missing from the boarding school for the day. A girl at school said that she would be friends with someone like me, who is living with HIV, but she would not accept a gift from me, because sharing can lead to harm. I cannot even share my clothes with my friends because I am embarrassed to tell them.
Figure 12Pills, food, and school.
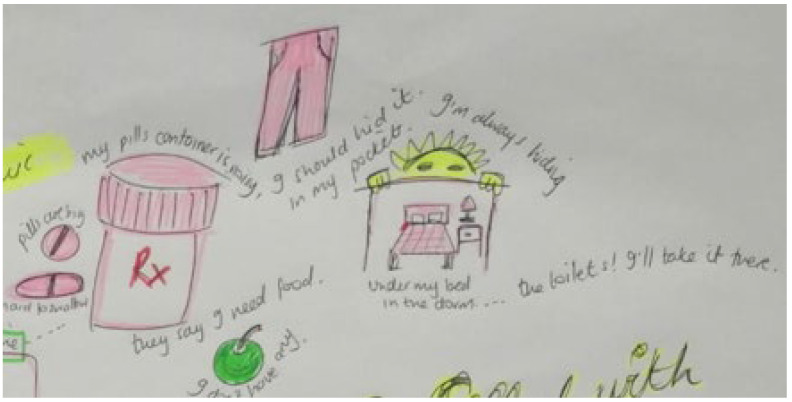


Storyline 7: Weighing up Health against Socio-Economic Constraints and Educational Priorities (Figure 13 and Figure 14)

I am always negotiating my health. We do not have money for food, which I must take with my medicine. People in my community think that we are bad children because we have HIV…that we make people drink and have sex with us. Even our family members treat other children in the house better than us. I must do extra chores, sweeping and mopping, while the others are doing their homework. People shout at us and use violence against us. We feel like a burden because going to the clinic costs money to travel and I still need school fees. They do not think I need to stay in school because I have this disease in my body. I am afraid of them. They are afraid of me. ‘Your mind is not settled; you think of many problems at the same time yet do not have solutions to these problems’. The community thinks we are bad and even if I try to ignore them, what they say affects me and how I feel about myself. It makes me want to not take my medicine. The doctors told me that having HIV is not the end of the world that I can study and get a degree and have a future like other people. Even though I do not like being across the table from the doctor at the clinic, I believe him and promise to try and take my medicine as I should from tomorrow.
Figure 13Economic and social constraints.
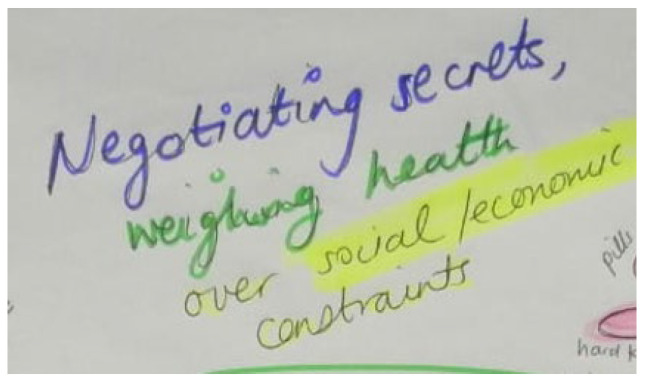

Figure 14Decisions, fear, and stigma.
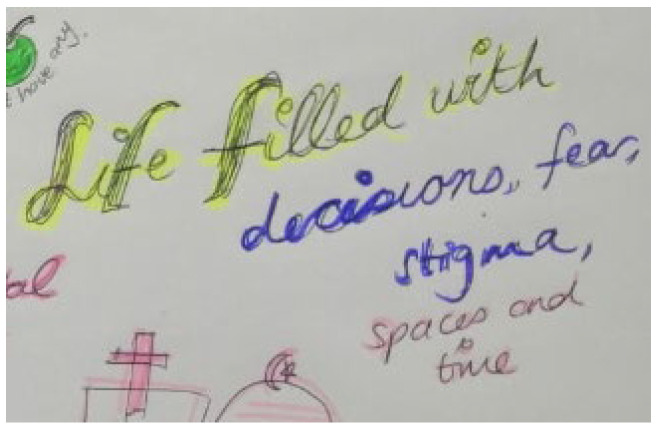


Storyline 8: Resilience, Motivation, and Future Goals Prompts Adherence as Habitual Behaviour (Figure 15)

Now that I am older, I understand that living with HIV won’t kill me if I take my medicine. We have better treatments now than my parents had. I have hope for my future, I want a career…a family…children…I won’t give up on my dreams. I see people from my community getting good jobs and having nice things. I also want those things. My family supports me to go to school and makes sure that I eat healthy food.
Figure 15Motivation to be healthy and adhere.
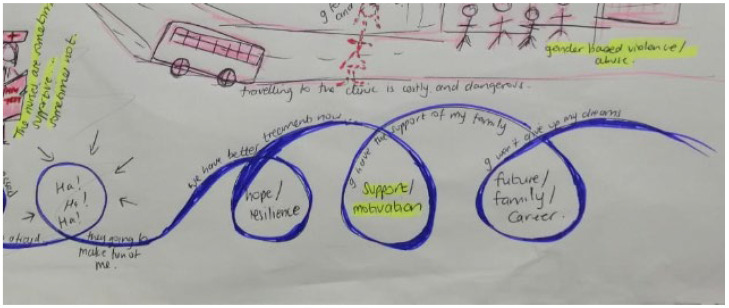


Storyline 9: Places and Spaces in the Community that Support and Threaten Adherence (Figure 16)

The community I live in has churches, mosques, and community halls. I could go see the pastor at my church and ask him for help, but what if he tells other people. Sometimes the church gives us food and I feel good after I attend services. We pray there. My friend goes to a church that has a support group. I walk past the church when I go to the clinic to collect my medicine. I feel like everyone is watching me. It is so far to walk to the clinic. When I take the bus or the taxi to the clinic, they will know where I am going. I am afraid they will judge me. There are men standing around and I must travel alone. I am scared. They will ask me where I am going so early in the morning. What if they see my pills in my bag? I do not have enough money to take transport home from the clinic. There is a park in my community where I can sit with my friends. It can also be dangerous at times when I am walking alone but I cannot ask anyone to go with me because they cannot know why I am going.
Figure 16Community places and spaces.
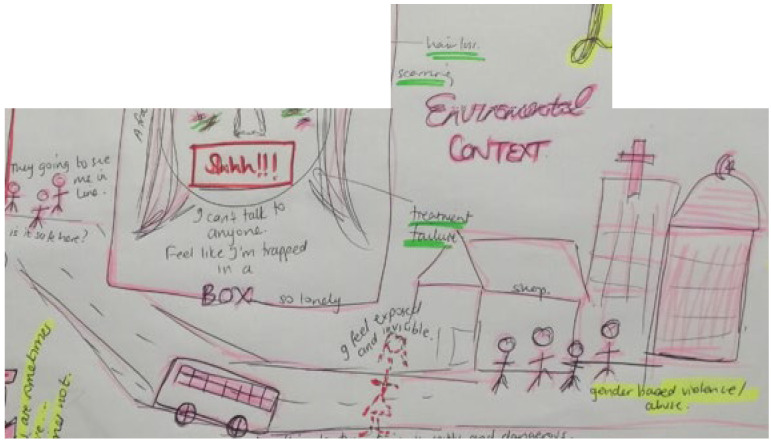


## 4. Discussion

The aim of this QES was to explore the interaction between individual, family, community, socio-cultural, health system, political, and material domains, in relation to adherence to ART for YLPHIV, from a new materialism perspective. The findings explore the assemblage of adherence for ALPHIV through storyboarding, highlight the practicalities of applying the new materialist perspective on biomedicine, and proposes a new model of adherence to ART for YLPHIV that emphasises the presence of material dimensions that it affects. 

In adopting the new materialist perspective, we advance the understanding of treatment adherence and biomedicine into a more transdisciplinary conceptualisation. This includes refocusing our efforts not only on the fixed relationships, causal factors, and outcomes but also on the exploration of agentic forces at play, at any given time and zooming out the frame of only the biological, psychological, and social dimensions onto other co-existing ones, such as structures, materials, and matter, for further exploration. Illness and health care decisions can be seen as a responsive performance in response to both molecular, societal, and material influences [19,25]. New materialism does more than add the material; it challenges the whole idea of systems and promotes a permanent process of becoming and doing in repetitive, responsive motion [18]. This review repositioned the patient as an agentic being amongst other agenetic human and non-human beings, which offered a different perspective to understanding health, and specifically to adherence to ART. 

The main finding is that adherence is a multisensorial experience in a multi-intra-acting agentic world. Our different understanding is that treatment adherence has less to do with humans’ preferences, motivations, needs, and dispositions and everything to do with how bodies, viruses, things, ideas, institutions, environments, social processes, and social structures assemble when the phenomenon of adherence emerges. What matters for health is the complex intra-active rhizomatic assemblage of things and species—material, nature, human, more-than-human, corporeal, and technological. YLPHIV grow up in households where one or both parents may have passed away due to complications related to HIV. They start out the childhood journey with normalised views of death, loss, and ill health. The first visit to the clinic for many YLPHIV was when they became aware of their surroundings and other people; this was apparent in the findings. YLPHIV realised that they were standing in long queues, had to miss school to attend the clinic, and may be in a crowded facility—in a special area of the clinic demarcated for HIV services. The environment affected the patients actively. Some YLPHIV reported that not all health care workers were supportive, but some were. The way that nurses engage with patients is important to their experience. The material markers such as their patient folder, the clinical room setting, the HIV test, the finger prick, and the nurses and doctors were identified. 

The way we tend to represent relationships in systematic reviews, as lines between theoretical concepts, requires a rethink to fully grasp the entangled nature of complex phenomena such as ART adherence. This QES approaches the findings through storyboarding as a means of synthesis to generate knowledge and insights [18,145]. A challenging step in this method was applying a new materialist lens to studies which did not prioritise materiality and then moving from the data extraction drawing to a more holistic storyboard.

### Strengths and Limitations of This Review

Following the conventions of the prospective protocol registration on PROSPERO, the wide comprehensive search, independent and in duplicate screening and selection of studies, as well as critical appraisal, including transparent reporting, we attest to the methodological strength of this review. To our knowledge, this QES is the first to adopt a new-materialist perspective and the first to introduce storyboarding as a method and an analytical technique that visualises the idea of material markers in a review. The strength of using new materialism as a theory is that it supports evidence synthesis through investigating primary studies to discover more and or different layers of a particular phenomenon. Conventional theoretical models of health have placed humans in a central position, when in fact, we have ignored the agency of our spaces and things (matter) around us. Herein lies the theoretical analytical strength of this review, which uses a transdisciplinary perspective to transition the human as the central figure to an agentic force amongst other forces. Adopting new materialism as a lens was challenging. We found a dominance of humanly oriented papers on the primary level, which meant that we analysed papers that in themselves pay little attention to material markers. Second, while review findings might be more digestible for uptake in the form of storyboards, it requires a basic level of artistic skills to create the drawings. However, using open-source pictures that closely resemble images of meaning can be a useful alternative. A recommendation to mitigate this challenge is careful planning of the author team to include this skill or investing in graphics and animation software. Finally, we had to be conscientious in ensuring that our own interpretations and biases did not overpower the overarching findings in the included papers. Through discussion within the author team, induplicate data extraction and analysis, as well as iterative drawings of the storyboard while including quotes directly from the paper supported the transparency of the process. 

## 5. Conclusions

In this review, we study the complex configuration of different agents that impact therapy adherence of YLPHIV. We applied the idea of assemblage and entanglement of the human with the non-human, natural, and cultural from the new materialism theory to a synthesis level. This QES explored not only what matters in the stories of the YPLHIV but also how matter comes to matter in studying adherence to ART. The main storylines identified in the primary studies materialised themselves in the drawing. These included navigating clinic visits, health care workers, and privacy; disclosure and psychological reactions to HIV diagnosis; grieving loss of caregivers and challenging familial relationships; bodily changes, internal stigma, and suicidal ideation; negotiating power, sex, and risky behaviours; fear of false judgement and stigma forces YLPHIV to keep their pills and status a secret; weighing up health against socio-economic constraints and educational priorities; resilience, motivation, and future goals prompt adherence as habitual behaviour; and places and spaces in the community that support and threaten adherence. This increased our understanding of the study results. In turn, the material markers illustrated in the drawing were presented through a collective voice in the first person. This style of writing and the use of visuals will most likely appeal to different publics, including citizens, societal stakeholders, practitioners, and policy makers. 

This paper does not only highlight what is important to consider in treatment adherence but also makes an important theoretical contribution to the field of systematically reviewing evidence. Further experimentation of the storyboarding technique as a synthesis approach could include testing the use of inductive or deductive coding in the process, mediums and modes of storyboarding, member checking of data with the authors of included studies, and multi-model perspectives on how to structure review findings. Additionally, we would like to issue a call to those in primary research, policymaking, and advocacy to start considering what matter comes to matter in studying complex health care problems. This means that addressing adherence to ART through policy, programmes, and interventions, a comprehensive approach, mindful of all elements and their influences on one another, must be considered. YLPHIV are already challenged by the developmental phase of adolescence, that coupled with the loss of parents, living with secrets, shame, and long-term mental health concerns, changes their interaction with their environments and the meaning ascribed to objects and spaces they encounter. Reciprocally, the objects and materials influence their experience of adherence and illness. Practitioners should rethink the patient as an isolated agent but as someone who is in a constant state of becoming within an assemblage and account for the conditions of everyday existence for those YPLHIV and the range of bodily and systemic factors which assemble their condition.

## Figures and Tables

**Figure 1 ijerph-19-11317-f001:**
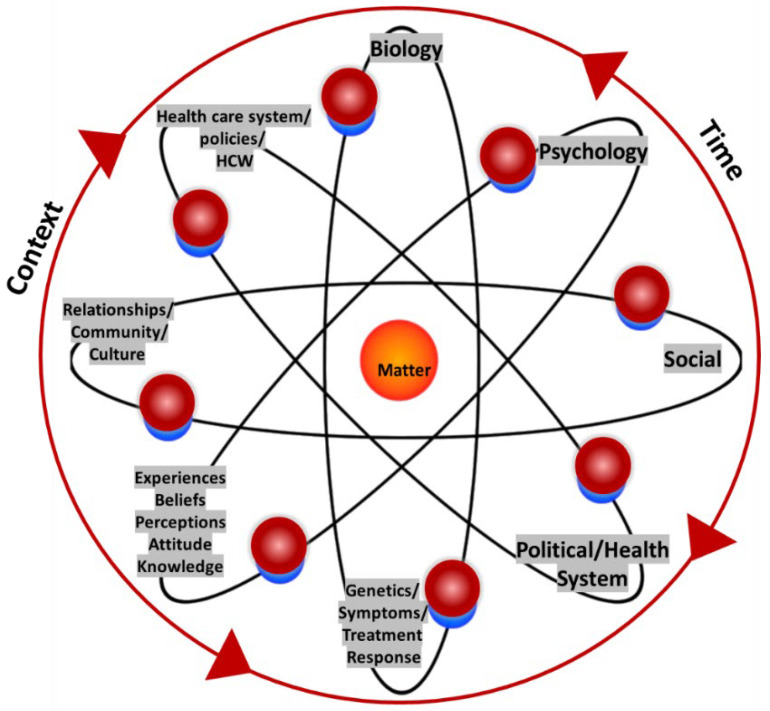
Biopsychosocial-material model of adherence.

**Figure 3 ijerph-19-11317-f003:**
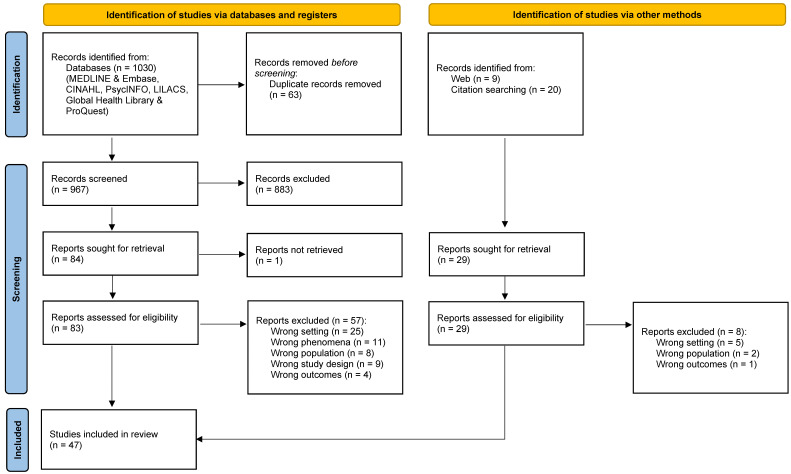
PRISMA Flowchart.

**Table 1 ijerph-19-11317-t001:** Themes, material markers, and storylines.

Dimension	Theme	Material Marker	Storyline
**Biological**	Changes in the body	Rashes, acne, hair loss, physically weaker or stronger	Bodily changes, internal stigma, and suicidal ideation
	Poverty and food insecurity	Food and money	Weighing up health against socio-economic constraints and educational priorities
	Biological sensations	Nausea, hard to swallow big tablets	Bodily changes, internal stigma, and suicidal ideation
	Treatment failure	Returning to the clinic, feeling sick and weak, visits to clinic or hospitalisation	Bodily changes, internal stigma, and suicidal ideation Navigating clinic visits, health care workers, and privacy
**Psychological**	Motivation, hope, and resilience	Condoms for protective sex, child/future children, desire for employment and ‘normal’ life	Resilience, motivation, and future goals prompts adherence as a habitual behaviour
	Internal stigma, depression, and suicidal ideation	Linked to physical appearance. Whole world feels like a box you are trapped inside Rat poison, throw myself into a lake, cut my neck and die, and feeling trapped in a box	Bodily changes, internal stigma, and suicidal ideation
	Fear of disclosure and secrets	Not being honest with friends by hiding pills inside tables, under pillows and beds, and in pants pockets. Noisy pill bottles and the colour of the pills.	Fear of false judgement and stigma forces YLPHIV to keep their pills and status a secret
	Knowing and fearing loss	Loss of parent/s or family members. Fear of losing romantic partners.	Grieving loss of caregivers and challenging familial relationships Fear of false judgement and stigma forces YLPHIV to keep their status a secret
**Socio-cultural**	Romantic relationships and intimate partner violence	Cheating with another person, condoms, protective sex, broken hearts, sperm, partner does not want to have sex, partner wants to have unprotected sex, gender-based violence/action of hitting, and words being said that are humiliating.	Negotiating power, sex, and risky behaviours
	Familial relationships	YLPHIV treated differently to siblings, food is separate, only one taking medication, additional chores (brooms, dishes, and laundry), no expectation to finish school, loss of parents.	Grieving loss of caregivers and challenging familial relationships
	Peer pressure and friendships	Hiding pills away from friends, staying at home when not feeling well.	Fear of false judgement and stigma forces YLPHIV to keep their status a secret
	Substance use and abuse	Alcohol and other substances, no condoms.	Negotiating power, sex, and risky behaviours
	Places in communities	Church or mosque linked to hope, bus stops, and taxi ranks linked to travel to clinic and work, busy street, and long roads to walk to the clinic, community members seeing YLPHIV at the clinic or waiting in a queue.	Navigating clinic visits, health care workers, and privacy Places and spaces in the community that support and threaten
	Socio-economic context	Low-income community, community violence—guns, gangsters standing on street corners and in the roads, no food to take medicine	Weighing up health against socio-economic constraints and educational priorities
**Health system-political**	Location of the clinic	May be too far to walk—need transport (bus, taxi, or car)—may be expensive (money)	Weighing up health against socio-economic constraints and educational priorities
	Health care workers and consultations	The test, accompanying family member, space in clinic may be crowded with no privacy, patient folder, the health care worker, doctors	Navigating clinic visits, health care workers, and privacy Disclosure and psychological reactions to HIV diagnosis
	The clinic	Information pamphlets, posters on the wall, all the patients in the clinic, the building and what it represents to the community, collecting medicines, place to go when sick	Navigating clinic visits, health care workers, and privacy Places and spaces in the community that support and threaten
	School	Choose going to school versus going to clinic, hiding pills inside table or under bed at boarding school. Hiding pills in pants pockets, going to the toilet to take it	Navigating clinic visits, health care workers, and privacy Fear of false judgement and stigma forces YLPHIV to keep their status a secret Weighing up health against socio-economic constraints and educational priorities

## Data Availability

Not applicable.

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
