# Peer review of "Storyboarding HIV Infected Young People’s Adherence to Antiretroviral Therapy in Lower- to Upper Middle-Income Countries: A New-Materialist Qualitative Evidence Synthesis"

_ijerph, 2022, doi:10.3390/ijerph191811317_

Round 1
Reviewer 1 Report
This article employs a qualitative evidence synthesis of storyboarding to explore HIV infected young people’s adherence to antiretroviral therapy in lower to upper middle-income countries. Authors assert that HIV is a global public health challenge. Furthermore, authors contend that young people living with perinatal infections of Human Immunodeficiency Virus (YLPHIV) face a chronic disease with treatment that includes adherence to life-long antiretroviral treatment (ART) and are a key population with limited representation in the literature. This article analyses and interprets findings of images of meaning, a storyboard, and storylines as it relates to young people’s adherence to antiretroviral therapy.
The review is as follows:
1. Line 14 and line 67 – Write out the acronym ‘BPS’ the first time it is introduced.
2. Lines 162-169 – The aim of the paper is not introduced until the end of the Introduction and four pages into the manuscript. Perhaps the aim of the paper could be introduced earlier and before line 127 (1.1. Why is it important to do the current review?).
3. Lines 128-129– In “Young people are especially vulnerable in South Africa due to reasons such as economic and educational inequality” this is a good mention. It would be important to mention the role of apartheid that contributes to disproportionate health inequalities among Black South African youth.
4. Line 268 – Good inclusion of an image on the synthesis of multi-modalities using storyboarding.
5. Good inclusion of Figure 3 in the PRISMA Flowchart.
6. Line 384 - Figure 6 of Storyboard of adherence to ART for YLPHIV is compelling. Overall, good inclusion of various insightful figures
Overall, this is a unique, insightful paper on a compelling topic. It is comprehensive, well-written and relevant. Attending to a few clarifying questions may help to improve the overall paper.
Author Response
We would like to thank the reviewer for taking the time to read our paper entitled, “Storyboarding HIV infected young people’s adherence to antiretroviral therapy in lower to upper middle-income countries: A new-materialist qualitative evidence synthesis”. Thank you for reading our manuscript with interest and for the valuable comments that were provided. We appreciated each comment and discussed it in the author team, with these in mind we revised our manuscript.
As recommended, we made major editorial changesAs suggested by reviewer 1, we moved the aim of the paper up in the introduction and provided clearer images for inclusion in the manuscript. We created a digital version of the storyboard and included it as image 7. Importantly, we provided more insight into the Apartheid legacy and how the livelihood of oppressed persons continued to be affected.
Please see the attachment,

Reviewer 2 Report
Thank you for the opportunity to do this review. This paper uses a materialist qualitative evidence synthesis to describe storyboarding HIV-infected young people’s adherence to antiretroviral therapy in lower to upper-middle-income countries.
This is an interesting article that describes bold research that I believe can interest other researchers. I liked the text and how the authors performed the research, and I think their findings can be helpful. Nevertheless, I think that some minor aspects could be revised:
The introduction is interesting, but it is a bit long. If possible, I would encourage the authors to shorten it. The overall article is long. I recommend the authors shorten it, if possible, to increase its readability. This is just a proposal.
The Analysis by the Storyboarding method is innovative and bold but can also confuse readers. For example, the authors describe that «All images and the storyboard were created through rough drawing and symbols, and then created digitally. The drawing had several iterations and followed a cyclical process of drawing, a discussion between authors, updating the drawing to reflect themes that were missing or represent further complexity in the mapping of images of meaning», and «The ‘story’ is our summative narration from which we will share the ‘story’ of adherence to ART for YLPHIV from a visual display that combines narrative interpretations with drawings using a new materialist lens». This methodology is a bit confusing and introduces a potential bias, as the authors interpret the data from their point of view. Perhaps these aspects could be better described in the Methods section and in the limitations section.
The Strengths and limitations section would fit better at the end of the discussion section and before the conclusions. This section perhaps should show more limitations regarding the storyboarding method, which is the essential feature of this interesting research.
Once more, the conclusions, albeit interesting, perhaps are a bit too long. I would recommend the authors shorten them, if possible, to clearly and succinctly show the main findings of their research.
Author Response
We would like to thank the reviewer for taking the time to read our paper entitled, “Storyboarding HIV infected young people’s adherence to antiretroviral therapy in lower to upper middle-income countries: A new-materialist qualitative evidence synthesis”. Thank you for reading our manuscript with interest and for the valuable comments that were provided. We appreciated each comment and discussed it in the author team, with these in mind we revised our manuscript. As recommended, we made major editorial changes
Reviewer 2 provides interesting insights into the application of our novel methodology, and we carefully reviewed the manuscript to see where we could clarify the steps we used. This resulted in us adapting the text and including a final step called ‘researcher reflexivity’ to make the reader aware of how researcher positionality influences the research process and analysis.
Please see attachment.

Reviewer 3 Report
COMMENTS TO THE EDITORS of International Journal of Environmental Research and Public Health
Reference. Manuscript ID: ijerph-1783122
The paper presented under the title "Storyboarding HIV infected young people’s adherence to antiretroviral therapy in lower to upper middle-income countries: A new-materialist qualitative evidence synthesis”
I consider that the objective is of great interest, however, I recommend incorporating an aspect that I highlight at the end of this report, and that I justify.
I think that the theme it addresses is important. The purpose of the investigation is described by the authors in lines 162-169:
The aim of this QES was to explore adherence to ART as an assemblage of different agents intra-acting with each other within the framework of the BPS model with a new materialist perspective. This QES will be the first to adopt the new-materialist perspective that “matter is to be studied not in terms of what it is, but in terms of what it does: what associations it makes as it affects and is affected, and what consequences derive from these affective intra-actions between agents” (21). It is also the first to introduce storyboarding as a method and an analytical technique that visualizes the idea of material markers in a review.
On lines 193-202, the authors describe the types of studies they used to do their research:
2.2.1. Types of studies
Studies using a qualitative study design, and qualitative methods for data collection and analysis were considered for this review. Eligible qualitative study designs included ethnographies, process evaluations, case studies, and mixed methods studies containing qualitative data. Qualitative data collection methods included observations, interviews, and focus groups. Qualitative data analysis methods including thematic analyses, narrative analyses, grounded theory, content analysis and descriptive presentations of findings were included in this synthesis. All quantitative studies, such as cohort, survey, cross sectional, randomized control trials, experimental and intervention studies, and other systematic reviews were excluded, as were purely descriptive papers such as opinion pieces and editorials.
In my opinion:
This research should be published for two reasons, one, because of its novelty, and another because of the topic it addresses, which is of great interest. However, I think that the authors should dedicate at least one paragraph to a relevant aspect that I am going to argue.
Treatment adherence is a highly relevant aspect for the good recovery of patients, to guarantee a reasonable quality of life, and so that their health status does not worsen if the disease is chronic and not curable. Thus, knowing which factors have an important influence on non-adherence to treatment is of paramount interest, and this aspect must be studied from all possible perspectives. This alternative way of studying what non-adherence to treatment depends on is novel and should be considered because it may shed light on this issue.
However, non-adherence to treatment has a significant negative influence on all types of research because it entails the loss of data registration, and thus very negatively affects the analysis of data from all research, this aspect being very relevant when you want to test a causal hypothesis about the effect of an intervention. For this reason, from my point of view, the authors should dedicate a paragraph to this aspect. I recommend that alluding to the importance of this, the authors cite two bibliographical references that faithfully reflect this aspect.
Vallejo Seco, G., Ato, M., Fernández García, M. P., & Livacic Rojas, P. E. (2018). Sample size estimation for heterogeneous growth curve models with attrition. Behavior Research Methods, 51 (3). DOI: 10.3758/S13428-018-1059-Y
Fernández-García, M. P., Vallejo-Seco, G., Livácic-Rojas, P., & Tuero-Herrero, E. (2018). The (ir) responsibility of (under) estimating missing data. Frontiers in Psychology, 9, 556. DOI: 10.3389/FPSYG.2018.00556
One of these bibliographical references deals with the subject from the care of the research design, and another, on the care in the planning of the sample size taking into account the loss of data. In this way, the authors will be able to reflect that they are aware of this problem although they do not address it in their research, and through the two references they holistically allude to this problem.
Author Response
We would like to thank the reviewer for taking the time to read our paper entitled, “Storyboarding HIV infected young people’s adherence to antiretroviral therapy in lower to upper middle-income countries: A new-materialist qualitative evidence synthesis”. Thank you for reading our manuscript with interest and for the valuable comments that were provided. We appreciated each comment and discussed it in the author team, with these in mind we revised our manuscript. We appreciate the implication of the novelty of our data analysis method.
Reviewer 3 challenged us with a great comment regarding the study of non-adherence and its implications on missing data. As this paper focuses on the assemblage, as related to new materialism, in a qualitative explorative way, we move away from the concepts of numbers and statistics. Studying the phenomenon of adherence, with its barriers and facilitators, contextual dynamics, experiences, and perceptions; we do not attempt to make any causal hypothesis.
Although we acknowledge the idea that numbers and narratives are part of the research assemblage or apparatus through which we try to understand a complex phenomenon, and we see value in the position of the review comment in a quantitative context, it moves us outside of the scope of the paper.

Round 2
Reviewer 3 Report
COMMENTS TO THE EDITORS of International Journal of Environmental Research and Public Health
Reference. Manuscript ID: ijerph-1783122
And comments to Authors of the paper presented under the title "Storyboarding HIV infected young people’s adherence to antiretroviral therapy in lower to upper middle-income countries: A new-materialist qualitative evidence synthesis”
Second revision
The very important assessment I made of this article has not been taken into account.
Please, take into account the problem of missing information, in this case due to data loss, and include some of the very important references I suggested. It is irresponsible not to.
I refer what I suggested in the first review I did:
In my opinion: the authors should dedicate at least one paragraph to a relevant aspect that I am going to argue.
Treatment adherence is a highly relevant aspect for the good recovery of patients, to guarantee a reasonable quality of life, and so that their health status does not worsen if the disease is chronic and not curable. Thus, knowing which factors have an important influence on non-adherence to treatment is of paramount interest, and this aspect must be studied from all possible perspectives. This alternative way of studying what non-adherence to treatment depends on is novel and should be considered because it may shed light on this issue.
However, non-adherence to treatment has a significant negative influence on all types of research because it entails the loss of data registration, and thus very negatively affects the analysis of data from all research, this aspect being very relevant when you want to test a causal hypothesis about the effect of an intervention. For this reason, from my point of view, the authors should dedicate a paragraph to this aspect. I recommend that alluding to the importance of this, the authors cite two bibliographical references that faithfully reflect this aspect.
Vallejo Seco, G., Ato, M., Fernández García, M. P., & Livacic Rojas, P. E. (2018). Sample size estimation for heterogeneous growth curve models with attrition. Behavior Research Methods, 51 (3). DOI: 10.3758/S13428-018-1059-Y
Fernández-García, M. P., Vallejo-Seco, G., Livácic-Rojas, P., & Tuero-Herrero, E. (2018). The (ir) responsibility of (under) estimating missing data. Frontiers in Psychology, 9, 556. DOI: 10.3389/FPSYG.2018.00556
One of these bibliographical references deals with the subject from the care of the research design, and another, on the care in the planning of the sample size taking into account the loss of data. In this way, the authors will be able to reflect that they are aware of this problem although they do not address it in their research, and through the two references they holistically allude to this problem.

Author Response
Dear Editor and Reviewer 3
We thank the reviewer for their comments and feedback. Following discussion we have resolved to include the paragraph below to address the comment received from the reviewer.
As this paper focuses on the assemblage, as related to new materialism, in a qualitative explorative way, we move away from the concepts of numbers and statistics. In studying the phenomenon of adherence, with its barriers and facilitators, contextual dynamics, experiences, and perceptions; we do not attempt to make any causal hypothesis. Adherence to treatment is a life saving measure for people living with HIV and ensures quality of life. Therefore, the study of non-adherence to treatment is of particular importance and interest. Implications of non-adherence are far reaching and affect the person, their families, as well as causing long term implications for health systems. In quantitative research using designs such as randomized control trials or case control studies, missing treatment doses can compromise the data quality and analysis of research (Fernández-García, 2018). In this qualitative evidence synthesis, we focus on non-adherence to treatment not as a measure of missed data, but rather as the inclusion of the voices, perspectives, and experiences, of missing groups or persons (Murris, 2020), specifically those of YLPHIV, from the literature and the narrative.
This paragraph is located on page 8 from line 182-195
Warmest wishes
The author team
